# Workplace Bullying as Experienced by Managers and How They Cope: A Qualitative Study of Swedish Managers

**DOI:** 10.3390/ijerph16234693

**Published:** 2019-11-25

**Authors:** Christina Björklund, Therese Hellman, Irene Jensen, Cecilia Åkerblom, Elisabeth Björk Brämberg

**Affiliations:** 1Karolinska Institute, Institute of Environmental Medicine, Unit of Intervention and Implementation Research for Worker Health, Box 210, SE 171 77 Stockholm, Sweden; irene.jensen@ki.se (I.J.); cecilia.akerblom@authenticpeople.se (C.Å.); elisabeth.bjork.bramberg@ki.se (E.B.B.); 2Uppsala University, Department of Medical Sciences, Occupational and Environmental Medicine, Dag Hammarskjölds väg 60, 752 37 Uppsala, Sweden; therese.hellman@medsci.uu.se

**Keywords:** workplace bullying, managers, qualitative study, work environment

## Abstract

*Background:* The aim of the study was to describe factors that contribute to the occurrence of workplace bullying, that enable it to continue and the coping strategies managers use when they are bullied. *Methods:* A qualitative study design was applied. Twenty-two managers from the private and public sectors were interviewed. Data were analyzed by means of content analysis. *Results*: Several factors could be linked to the bullying: being new in the managerial position; lack of clarity about roles and expectations; taking over a work group with ongoing conflicts; reorganizations. The bullying usually lasted for quite some time. Factors that allowed the bullying to continue were passive bystanders and the bullies receiving support from higher management. The managers in this study adopted a variety of problem-focused and emotion-focused coping strategies. However, in the end most chose to leave the organization. Only four remained in their managerial position at the time of the interview. *Conclusions:* The study concludes that bullying can occur in all types of organization. To prevent it we need to look primarily at organizational factors. Social support is also crucial for managers’ ability to cope successfully with bullying.

## 1. Introduction 

Workplace bullying has been the subject of international research for several decades. This has given us a better understanding of its pervasive and detrimental effect on all levels [1]. In fact, bullying is now considered one of the most detrimental stressors in contemporary working life [2]. 

Bullying is defined as the “systematic mistreatment of a subordinate, a colleague, or a superior” [3] (p. 3). It thus includes the phenomenon of a manager being a target of bullying. In this study we used the definition of workplace bullying developed by Einarsen et al., 2011 [4], according to which it is related to situations in which an individual is harassed, offended, or socially excluded and in which their work is negatively affected. The bullying must also occur repeatedly and regularly over a period of time and must be an escalating process. Power disparity between perpetrator and target is an important characteristic of bullying [4]. Workplace bullying is therefore often associated with managers harassing their staff. This is because workers are perceived as vulnerable, while those in positions of power or authority, such as managers, are seen as perpetrators [5]. However, anyone in an organization, irrespective of position, can be the target of bullying. The bullying of managers is often referred to as upward bullying and was first discussed around the turn of the 21st century [6].

Previous studies have identified a prevalence rate of between 10% and 20% of managers reporting that they have been bullied, primarily by their employees [7]. When a manager is bullied, especially by subordinates, it reflects a situation in which those who are lower down the hierarchy gain power from factors that are not related to their position. These factors can include the control of information [8], expertise [9,10], and referent power [10]. 

One of the most extensive qualitative studies was conducted on bullied managers in the Australian public sector. It found that work environment, organizational changes, and power were the factors which contributed to upward bullying. Upward bullying refers to subordinates bullying superiors in situations where there is a power difference [11]. The study also showed that managers were more vulnerable to bullying at times of insecurity in the organization, for example when a manager was new to his or her managerial position, and in organizations where the manager was highly dependent on his or her staff, i.e., when there was a reversal of power relationships [12]. In addition, a recent overview of bullied managers identified several work-environment factors as contributing to upward bullying, namely role overload, competition, change, and isolation. These findings indicate that a subordinate would need to augment their power with support from other person(s) in authority (i.e., other manager(s) and supervisor(s)) to bully upwards and overcome the positional power of a manager [7]. D’Cruz and Rayner (2013) call this “cross-level co-bullying”—in other words bullying that involves individuals at different hierarchal levels within an organization [13] (p. 597).

A number of behaviors have been identified as related to managers being bullied. This type of bullying tends to be fairly subtle in the early stages, possibly due to the risks involved in more open displays of negative behavior towards a person in authority [14]. Branch (2007), for example, identified a combination of covert behaviors in the early stages, such as withdrawal of information or knowledge, that then became more overt, possibly due to an increased sense of power on the part of the employee [12].

In a later publication, Branch et al., 2018 point out that, while general workplace bullying and the bullying of managers have much in common, many processes in the latter focus on the use and abuse of power towards those in a higher hierarchical position [7]. Authors of a recent qualitative study into bullied managers and power identify a “power cycle” [15] (p. 21), according to which the use of power tactics by others can be triggered when the legitimate power of managers is undermined. This often begins covertly. Then, as the power of the perpetrator(s) gets stronger (or the manager is worn down), power-change tactics can be used. These tactics tend to be more overt and direct. The cycle then feeds back into further loss of legitimate power by the manager and a continuing cycle of escalating tactics [15].

### 1.1. Coping with Workplace Bullying

Being exposed to bullying is an extreme stressor which involves the targeted individual, the bully and surrounding persons, for example in the workplace. Individuals have a natural tendency to find ways to deal with or at least minimize the impact of stressful events.

Individuals exposed to bullying or stress use a variety of strategies to cope with the situation [16]. Folkman and Lazarus (1980) identified two types of coping, namely problem-focused coping (where the focus is on managing the encounter) and emotion-focused coping (where the focus is on regulating the emotion). In many cases, an individual under stress will use both types of coping strategy [17]. The effectiveness of a particular strategy may vary, while the choice of strategy may also reflect the severity of the bullying and the psychological state of the target. Lazarus and Folkman (1984) argue that an individual in a stressful situation will evaluate the resources that are available to him or her (coping resources), which could either be internal resources (for example personality, locus of control, etc.) or external (job-related) ones [18]. A fairly recent systematic review identified several factors as coping resources, for example support from co-workers and managers and self-efficacy, that may influence coping strategies [19]. Coping resources were found to be positively related to problem-focused coping strategies and negatively related to emotion-focused coping strategies.

It has been suggested that there are several stages to find coping mechanisms or dealing with workplace bullying. In the first phase, the target underestimates and avoids the problem. In the second, they lose patience and confront the situation. As time passes and they perceive a threat to their health, they seek support. When that support is not forthcoming, they are overcome by a sense of despair and move into a stage of destructive coping. In the final phase, the target of bullying frequently gives up attempts to resolve the situation and leaves the organization [20]. Similar processes have been identified by other studies [21].

To sum up, although research into workplace bullying has been carried out since the early 1990s, most studies have had quantitative designs [1]. In addition, previous studies have primarily focused on workers rather than managers [12]. In other words, there is only a small body of research into bullying from the perspective of the bullied manager [7,12].

Moreover, to the best of the authors’ knowledge, previous qualitative studies on bullied managers have been conducted in other countries than in Sweden. Lately, there has been a growing interest in cross-cultural perspectives and workplace bullying. Reasons for this include the identifiable social differences across cultures, including power distance, which can make a difference in workplace dynamics [12]. Therefore, to study bullying in another context could increase the knowledge in this area.

In the present study, we focus on factors that contribute to workplace bullying, those that enable the bullying to continue and the coping strategies managers use when they find themselves being bullied.

### 1.2. Aim of the Study

The aim of the study was to describe factors that contribute to the occurrence of workplace bullying, that enable it to continue and the coping strategies managers use when they are bullied.

## 2. Method

The present study has a qualitative design. Qualitative methods for data collection and analysis are well suited to exploring a contemporary phenomenon such as bullying in a real-life setting. This manuscript follows the recommendations of Tong and colleagues [22].

### 2.1. Participants

The selection of participants was guided by a strategic sampling procedure to get a variation in participant experience of the questions under study among the managers. In order to apply the strategic sampling, the following inclusion criteria were used: (1) identifying oneself as having been a target of bullying for approximately six months; (2) having responsibility for staff; (3) being proficient in Swedish or English. The definition of bullying referring to situations in which an individual is harassed, offended, or socially excluded and in which their work is negatively affected. The bullying must also occur repeatedly and regularly and over a period of time and be an escalating process. Power disparity between perpetrator and target is an important characteristic of bullying [4]. The criterion of approximately six months has been used in prior studies in order to differentiate between exposure to social stress at work and victimization from bullying, e.g., [23].

At the start of the recruitment process we distributed information through conventional and social media. The project was also presented at conferences attended by the human resources departments of a number of organizations. We also contacted representatives of unions that primarily represented managers. The recruitment process resulted in a total of 27 participants. Nineteen of these initiated contact with the project leader by e-mail, phone, or personal contact at lectures. Four participants were recruited by snowball sampling; two via human resources departments at their workplaces and two via their trade unions. Twenty-seven persons agreed to participate in the study. No one withdrew their consent.

Given the sensitivity of the issue, it could be difficult to ascertain whether the bullying was part of an interpersonal conflict (e.g., reprisal), or whether it counted as “real” bullying. We therefore gave all managers who were willing to participate in the study our definition of bullying. All managers who perceived themselves as being a target of bullying in accordance with this definition were included in the study. In addition, one of the first questions in the interview guide addressed the escalating progress of bullying (starting point, how long it lasted, coping strategies, and type of negative behavior). After the interviews had been performed and transcribed verbatim, all transcripts were read through and the participant’s description of the bullying was compared with the definition. Of the 27 participants, five managers (three women and two men) were excluded because their descriptions did not fulfil the criteria for bullying. The average age of the excluded managers was 59 years (ranging from 40 to 73 years of age); all five were employed in publicly funded organizations (municipalities or the government sector). They were excluded for the following reasons: one of the participants had witnessed but not experienced bullying him/her self; one did not have any staff responsibilities; one was being bullied by individuals outside the organization; two were dissatisfied with their working situation in general but were not experiencing bullying.

Seventeen of the participants were women and five were men; the average age was 54. Table 1 presents an overview of participants’ characteristics. The public sector participants were from county councils, the government sector or municipalities. The private sector participants represented the insurance, financial, and media sectors, as well as the process industry. 

### 2.2. Ethical Considerations

The Regional Ethics Review Board in Stockholm approved the study (Reg. no. 2015/2204-31/5). All participants were informed in advance that participation in the study was voluntary, that they could withdraw at any time without giving any reason and that it would not be possible to identify them when the findings were reported. Written informed consent was obtained from all participants.

### 2.3. Data Collection

Data was collected by means of semi-structured individual interviews. The qualitative data collection was based on the interplay between the interviewer and the participant, by means of the interviewer’s awareness of what the participant said during the interview, how the participant presented his/her experiences, and the interviewer’s ability to further explore and ask probing questions to elaborate and clarify ‘the interviewee’s individual experiences and thoughts [24].

All interviews started with background characteristics (age, gender, workplace, and sector). The participant was then asked to describe when the bullying started, how long it continued and what kind of negative behavior he/she had experienced. The topics that guided the interview were related to the escalating process of bullying and covered: (1) the start of the bullying (e.g., “Please talk about when you realized that you were the target of bullying.”; “Can you please describe the work climate and workload at your workplace at the time when the bullying started?”); (2) the period of full-blown bullying (e.g., “Can you tell me about an ordinary day at work at the time when you were the target of full-blown bullying?”; “How did the bullying affect you outside work?”); and (3) when the bullying stopped (e.g., “Please describe how the bullying ended.”). Follow-up questions (for example “Could you please tell me more about that situation?”) were asked during the interviews when appropriate.

The interview guide was tested in the first interview; no changes were made to the guide after the first interview, which was included in the data set. The participants were interviewed on one single occasion. Fourteen of the interviews were conducted by phone, two by Skype, two in the respondents’ homes and nine at the office. The average duration of the interviews was 90 min, with a range between 39 and 140 min. The interviews were conducted by C.B. and T.H., two female researchers. C.B. is an associate professor in economic psychology and T.H. an occupational therapist. C.B. and T.H. have doctorates and are trained and experienced in qualitative interviewing. C.B. and T.H. introduced themselves to the interviewees as researchers and gave their respective professions. All interviews were digitally recorded and transcribed verbatim by a transcription agency.

### 2.4. Data Analysis

The data (recordings and transcriptions) were cross-checked for accuracy by C.Å. and E.B.B. E.B.B. had chief responsibility for the data analysis, in close collaboration with C.B. and C.Å. The data was analyzed by means of qualitative content analysis [25] which aimed to explore the transcribed interviews with focus on their content. The analysis was performed in the following steps: (1) The transcribed interviews were carefully read through to gain an overall understanding of the content. (2) Sentences with content that related to the study’s objective were identified and marked (i.e., meaning units). (3) The meaning units from each interview were summarized. (4) The summarized meaning units (i.e., condensed meaning units) were labelled with codes. (5). All meaning units, condensed meaning units and codes for each interview were transferred to a Microsoft Word document. (6) The codes of all the interviews were compared with each other and those reflecting similar content were assigned to a sub-category. (7) Each sub-category was illustrated by a short description.

After Step 6 it became evident that some of the sub-categories were related to how the participants coped with being bullied. The definition of coping used was “cognitive and behavioral efforts to manage specific external and/or internal demands that are appraised as taxing or exceeding the resources of the person.” [18] (p. 141). Coping was then divided into two sub-categories and were analyzed in accordance with Lazarus and Folkman’s (1980) description of coping strategies and their distinction between problem-focused and emotion-focused strategies [17].

The sub-categories were compared and assigned to categories according to content. In the latter phase, C.B., T.H., and C.Å. commented on the analysis and the descriptions were adjusted after discussion among the authors. Thereafter, all authors reviewed the analysis and agreed on the final version. The analysis process is shown in Table 2. Two examples of short quotations from the interviews are used in the presentation of the findings in order to demonstrate the link between the data and the analysis.

## 3. Findings

The bullying of managers is part of a complex social exchange relationship. We saw that, given the formal power held by a manager, bullying could be used as a weapon against the management to voice employees’ dissatisfaction with organizational changes or other matters. This study identified a variety of types of perpetrator. In some cases, the perpetrator was a non-managerial employee, in others it was a senior colleague or a senior manager. There were also situations in which more than one party was involved indirectly or directly in the bullying.

The interviewees described a pattern according to which initial periods of bullying were followed by an escalating occurrence of active and passive incidents in the form of exclusion and more or less open threats. These could target either the individual or his/her work. The respondents reported that the bullying affected their mental health and gave rise to sleep disturbance, problems with concentration, and emotional reactions such as depression and anxiety. They also described their strategies for coping with the bullying at the workplace.

The findings are presented as follows: (1) Factors contributing to the occurrence of bullying. (2) Factors which enable it to continue. (3) Strategies used to cope with workplace bullying.

An overview of the results is presented in Table 3.

### 3.1. Factors Contributing to the Occurrence of Bullying

The following factors were identified: (1) being new to the managerial role or having recently returned from parental or sick leave; (2) having unclear roles and responsibilities; (3) taking over responsibility for work groups in which there were pre-existing interpersonal conflicts; and (4) being involved in a workplace reorganization.

#### 3.1.1. Being New to the Managerial Role or Having Recently Returned from Parental or Sick Leave

The bullying tended to start when the interviewees were either new as managers in the organization or after a period of sick leave or parental leave (i.e., the interviewee had been away from the workplace for several months). They perceived themselves as being in a vulnerable situation quite early in their new position, even if they all had previous experience of managerial roles, or on returning to work after an extended period of leave. They felt that it was the position itself that was being targeted for conflict, rather than they as individuals. One interviewee described how the bullying started:


*What really triggered such a terribly bad atmosphere among so many of the staff, even though we’d been working there for two years by then, was that I’d been away for a week because of a difficult family situation…*


#### 3.1.2. Having Unclear Roles and Responsibilities

According to the interviewees, the respective roles and responsibilities of workers and managers were not clearly defined, a factor that triggered the escalation. There was a lack of clarity about the manager’s role as well as about the responsibilities and work tasks of workers and managers. They also described unclear organizational structures which created uncertainty and ambiguity about their leadership. According to the managers, it was this ambiguity which gave rise to the bullying:


*And it got very awkward for me. X was a program coordinator; she used my staff as a staff function for her work. I mean, I don’t think it was wrong, but as the head of the unit (formal manager), I had absolutely zero influence.*


#### 3.1.3. Taking Over Responsibility for Work Groups in Which There Were Pre-Existing Interpersonal Conflicts

The interviewees described how they, as new managers, had to deal with interpersonal conflicts in their work groups. The conflicts had been escalating for some time and unsuccessful efforts to solve them had already been made. The interviewees therefore found themselves in a situation in which, as part of their job description, they were responsible for solving longstanding conflicts. These in turn were related to non-functional teams and workers voicing their frustration with organizational issues involving workers and managers.


*My boss knew about it, I asked him “Why haven’t you done anything about it? I’ve found a member of staff who’s been on sick leave for two years because of harassment at work and you haven’t even drawn up a rehabilitation plan after three months” This person told me “I can’t come back because of my co-workers.” My boss told me to get it sorted out.*


#### 3.1.4. Being Involved in a Workplace Reorganization

Workplace reorganizations were another factor perceived as a trigger for bullying. These could take the form of organizational changes; the amalgamation of two work groups; the introduction of a new level of management; or high staff turnover. Reorganization, particularly with regard to central issues such as downsizing and redundancies, could cause uncertainty. It was perceived as a threat to the work group as a whole. The interviewees described organizational changes with which both workers and managers were dissatisfied. For example, not receiving enough information about a reorganization caused a great deal of stress and anxiety in the work groups for which the managers were responsible. One of the interviewees described the situation as follows.


*There were two groups, mine worked fine, the other didn’t. My boss thought it could be solved by merging the two groups and gave me the task of doing it. We got into a situation where the manager of the other group didn’t get on with our boss. I think our boss wanted to do two things at once. Get rid of the other manager, merge the groups and solve the problem inside the group. We both applied for the job and I got it and he could never understand why. He had to stay in the group. That’s when everything started, because he was so angry with me and got some people in the group on his side.*


### 3.2. Factors Which Enabled Bullying to Continue

The factors that were identified in the interviews were: (1) bystander behavior; and (2) support from the higher management for the perpetrators, bystanders, and scapegoating.

#### 3.2.1. Bystander Behavior

Being the target of bullying meant being the target of a collective process in which the managers were successively undervalued and criticized by staff and by other managers. The collective process formed the basis for the bystanders’ behavior. The bystanders did not participate actively in the bullying but were described as taking part by means of subtle behaviors such as spreading gossip or rumors about the manager who was being bullied.


*Loads of people saw it, they saw it at management group meetings, saw his behavior and how he made fun of me, but at the same time, they were a little bit vulnerable themselves too. They didn’t dare say anything, they couldn’t stand up for me, that’s what they said, some of them, after the meeting “that was terribly unfair, what he did”, quite passive, but they did see it.*


#### 3.2.2. Higher-Level Management Support for the Perpetrators, Bystanders and Scapegoating

Many of the interviewees stated clearly that higher-level managers were involved in the bullying in some way, either actively or passively. Where one or several employees were perpetrators, the interviewees reported situations in which members of the work groups sought support from managers at a higher level than the interviewee. In some cases, higher level managers began to act differently towards the interviewees and supported those employees who were doing the bullying. However, in most situations the higher-level managers were described as not acting at all. They thus became bystanders and, in some cases, even participated in the scapegoating of the vulnerable managers.


*… my management colleague, who is a man, ridicules me at management meetings, “my dear, you can’t understand this, you can’t possibly have an opinion about this.” The CEO saw this going on but didn’t do anything. I think that’s been the most irritating aspect of the whole thing.*


The interviewees also described situations in which their superior managers and the management team, and sometimes also the HR department and union representatives, blamed them for workplace problems that had already existed before they became manager for the work group in question. Finding a scapegoat, according to the interviewees, was a way for higher-level managers and the management group to do “something” about a work-environment problem without addressing the psychosocial workplace issues that were likely to require greater effort.

### 3.3. Strategies Used for Coping with Workplace Bullying

The coping strategies reported by the interviewees were used as attempts to minimize the negative outcomes of their experiences. Lazarus and Folkman’s (1980) description of coping strategies and their distinction between problem-focused and emotion-focused strategies has guided how we identified and categorized the interviewees’ appraisal of the bullying and their coping responses [17].

#### 3.3.1. Problem-Focused Coping Strategies

The problem-focused coping strategies identified in the interviews were social coping, confrontation, and “using the manager mandate”.

Social coping: seeking support from the organization

The interviewees reported using social coping to seek support from managers, human resources personnel and/or occupational health services to deal with and solve the problems at the workplace that they had identified as causing or contributing to the bullying. As a first step, the interviewees had told their managers about the situation and asked for support. As a second step, human resources became involved. The interviewees had approached the problem by asking for help in clearly defining their own professional role and responsibilities as well those of the employees.


*I realized that I had to do something about this. I contacted the personnel manager to say that I had now taken this action. To get help with dealing with the situation.*


The interviewees reported that receiving support from their immediate manager or the management group was a crucial factor in improving the situation. In one of the cases, the interviewee was able to dismiss the person who was bullying him/her, with the support of the management group.

The interviews revealed, however, that not all the managers had received the support they requested. Some interviewees had been ignored when they sought support or had been told that it was part of their supervisory role to deal with problems and conflicts. However, in a few cases the bullying had ended or improved as a result of extensive support from the immediate manager or higher-level management.

Social coping: seeking support from colleagues

The interviewed managers stressed the importance of support from colleagues. In some situations, it was crucial for improving the situation. One described how colleagues stood up for her, which resulted in the perpetrator having to leave and the bullying ceasing.

Confrontation

Confronting the person they had identified as the perpetrator was a strategy reported by the interviewees. The confrontation was described as a planned “occasion”, often with their manager or someone else present. Some of the interviewees reported that they had documented negative acts before the confrontation and then used them at the confrontation. For one of the bullied managers, the confrontation resulted in the bully leaving the company.
… after I went to the owners of the company and told them,” if you want me to stay then this person has to go, otherwise I’ll go”, I said. They wanted me to stay. I and one of the owners called the employee to a meeting and said that this person had to leave the company.
The use of their power/mandate as a manager

The interviewees described how they made use of the power/mandate that they had been given as managers. Thus, for example, some of the interviewees had decided to form new working groups in order to create a better work climate. Another use of this strategy was to move one or more persons from their usual duties to others. The aim of this was to prevent certain persons from working together, and in that way to limit opportunities for employees to support the colleague who was leading the bullying. The interviewees also described how they developed written action plans to deal with the bullying and other problems at the workplace.

#### 3.3.2. Emotion-Focused Coping

The interviewees also reported using emotion-focused coping strategies. The ways they dealt with their emotions when they were bullied can be categorized as social, solitary and avoidance coping.

Social coping: seeking support from family, friends or social networks

The interviewees mentioned using social coping, e.g., seeking support from family, friends or social networks outside work. The support of others gave the interviewees the opportunity to discuss the negative incidents and problems at the workplace. These discussions helped the interviewees to identify the underlying patterns that might explain the bullying, such as the presence of informal leaders, dysfunctional working groups, or interpersonal conflicts. This was important because it allowed the interviewees to step back from the bullying and realize that it was not related to them as individuals.
I had support, it was my family. That was all that kept me going, that’s the psychological support I needed.
Solitary coping

The interviewees described how distracting themselves with work was another strategy they used to handle negative emotions. This involved focusing on work and disassociating themselves from the surrounding problems. However, there were both pros and cons to this strategy: it enabled them to create a little more (emotional) distance between themselves and the bullying while at the same time they described themselves as becoming isolated and distanced from those employees who were not taking part in the bullying.
Avoidance coping

Another strategy the interviewees used was avoidance. They stopped socializing at work, both with the bullies and the bystanders. They stopped going to lunch, conferences and other social events with their employees. Avoidance coping could also be an emotion-focused act, described as changing one’s approach to interacting with others. By using an avoidance strategy, they protected themselves from the bullying and negative acts; the negative consequence of this was that they became isolated from social relationships at work.

#### 3.3.3. Leaving the Organization

At the time of the interviews, only four of the interviewees remained in their managerial positions. The others had left their positions and organizations as a consequence of the workplace bullying, which is a very common coping strategy. Some of them had been on sick leave caused mainly by stress-related diagnoses during the period immediately before they decided to leave. Some of those who left were offered a financial incentive to leave the organization and support by their trade union in the process. The managers accepted the offer of being bought out because it was impossible resolve the situation in any other way.


*I was called in to my senior manager… a poor girl from human resources was there too, and I was simply asked to quit. And I asked for the reason. he said that is was a collective decision. Then he wanted me to basically take my things and go as soon as possible and say nothing to the employees. Because not everyone had a bad opinion of me …*


The four who had remained in their positions had support either from colleagues or from a senior manager. In one of the cases, the person responsible for the bullying had left the organization.

## 4. Discussion

The aim of the study was to describe factors which contribute to the occurrence and continuation of workplace bullying and what coping strategies managers use when they are being bullied.

The findings revealed that being new in the managerial position, taking over work groups with pre-existing interpersonal conflicts, lack of clarity about expectations and roles, and re-organizations were factors behind the occurrence of bullying. Bystanders’ behavior and higher management’s support for the perpetrators allowed the situation to continue. Finally, the interviewees used a variety of strategies to cope with the bullying, including problem-focused and emotion-focused coping strategies. In many cases the only way to stop the bullying was to leave the organization or to receive the support of higher-level management or colleagues.

When asked about what gave rise to the bullying, most interviewees responded that it started when they were new as managers in the organization or after a period of sick leave or parental leave. This supports Branch et al.’s study of Australian managers [26].

Our findings reveal a lack of clarity about roles, goals, and responsibilities. This is in line with previous studies of bullying in the general working population which have identified role conflicts, role ambiguity, and lack of clearly defined roles as features of work organizations characterized by bullying [23].

Further findings revealed that several of the interviewed managers had taken over responsibility for work groups with pre-existing interpersonal conflicts. In such situations, there is a high risk that the new manager, instead of solving the ongoing conflict, will become the target of bullying. The manager is not part of the conflict from the beginning, but since the situation involves intense emotions which give rise to fear, suspicion, resentment, contempt and anger, it is very easy for the manager to become the target of bullying [27].

Changes and reorganizations are well-known trigger factors for bullying and were identified as such in the present study. This was also confirmed by the interviewed managers. Hogg and Therry (2000) suggested that organizational changes can be perceived by staff as a threat [28]. This threat can create a level of uncertainty which in turn can cause individuals to act against their manager [29].

The present study also examined factors which enable bullying to continue. According to the findings, such factors were linked to bystander behaviors as well as higher-level management supporting the perpetrators, bystanders, and scapegoating. Stress in an organization can increase the tendency to apportion blame, and as the process escalates, managers become scapegoats. The interviewees perceived themselves as breaking the norms and values of the work group and as being blamed for this.

We found that support for the perpetrators from a higher manager undermined the bullied manager’s authority—a factor which perpetuates the bullying. This finding is supported by a previous study [11]. This type of support is necessary to bypass the formal organizational position of the manager. The absence of supervisor support for the manager undermines his or her actual legitimate power and, even more importantly, the perception of his or her legitimate power. Lack of power is probably one of the most important factors implicated in the bullying of managers. In this study we found that several of the managers were harassed both by employees and higher managers—a situation which is extremely difficult to deal with. The term “cross-level co-bullying” is used to describe bullying which involves a number of hierarchical levels within an organization [13].

The managers in the present study described a variety of coping strategies. Previous studies of coping indicate that most targets usually try to stop the bullying by using a variety of strategies. In an interview study of how people cope with being bullied it was found that all the interviewees initially used problem-solving strategies. If this did not help, they resorted to other strategies at different stages of the process. Many of the managers in this study reported that the bullying affected their mental health and gave rise to sleep disturbance, problems with concentration, and emotional reactions such as depression and anxiety. The relationship between bullying and mental ill-health has been reported in many previous studies [30].

In the end, a fair number choose to leave [31]. In the present study, this is exactly what most of the participants ended up doing. Previous studies have shown that this is a common strategy [32]. Many interviewees unsuccessfully sought support from the organization. This is in line with the findings of previous studies [12]. However, the bullying often improved if the target received support from higher management. In fact, according to Houg and Dofradottir (2001), it is rare for a targeted manager to succeed in stopping the bullying without the help of others [33].

In this study we found that although the study was conducted in another country such as for example Australia, similar factors contributed to workplace bullying among managers. Being a manager seems to be a very specific role regardless of the setting. The present study strengthens our understanding of the specific factors that give rise to managers being bullied. It also contributes to our understanding of those factors that enable the bullying to continue. We hope that this better awareness will improve our understanding of how a bullying situation can be stopped rather than being allowed to continue. The study also identifies which coping strategies are used by managers, which in turn can help us to understand how to give better support to managers who are being bullied.

### Methodological Considerations

A strength of the present study is that the data reflects a variety of perspectives. Both men and women took part in the interviews and they represented both public and private sectors. All held managerial positions. Another strength of the study is its recruitment process. Given the sensitivity of the topic, we recruited interviewees by spreading information about the project via unions, human resources departments, social media, and the lecturers working in the project leader’s department. Twenty-seven persons were willing to participate in the study and were interviewed. Five interviews were excluded because their experience did not fulfil the criteria for bullying. None of the interviewees withdrew their participation, even though the interviews could well have brought back painful memories for them.

To ensure the integrity of the data analysis, the meaning units, codes, and categories were kept close to the data. Throughout the process of data analysis, the researchers compared the interviews, the meaning units, and codes in several steps. All the researchers contributed to the conceptual creation of the findings and the emerging results were continuously discussed by the inter-professional research group in order to improve rigor [25].

This study has several limitations. Because the recruitment process was not carried out by the research group it is not possible to establish the exact number of non-participants. The sample contains only a small number of men, which might be a limitation. Information about the managers’ length of managerial experience is lacking, which might also be a limitation. In all other ways, however, the strategic sampling method was satisfactory. To the best of the authors’ knowledge there is a lack of validated interview questions addressing bullying. For this reason, the authors themselves developed the questions used in the interviews and pilot-tested them in the first interview. The findings reveal the experiences of participants (managers, predominantly women) in a specifically Swedish setting, which might affect the study’s generalizability to other countries.

## 5. Conclusions and Implications

To conclude, the consequences of bullying may be especially severe when the target is a manager. It has been shown that in such cases, bullying can degenerate into abuse of escalating severity more rapidly than with other forms of workplace bullying [34]. Branch et al. (2018) identified several factors which can help to prevent the bullying of managers. Firstly, it is important to understand how the work environment can be used to bully another person. Secondly, managers must receive the training and development they need to perform their roles. Thirdly, managers must be aware of their power, which they can use to build a positive culture. Finally, senior management must to be ready to respond to the needs of managers and supervisors when they seek help, rather than dismissing their concerns as trivial or minor [7]. Several studies have reported that managers can be targets of bullying. In most organizations, grievance/complaints systems are more suited for employees in general. There is a constant need to review systems, not only to reduce the risk of abuse but also to evaluate how well the system works for managers. It is therefore important to investigate the occurrence of bullying among managers in order to enable organizations to build support systems as well as grievance/complaints systems to help the latter cope with situations that can often become intolerable.

## Figures and Tables

**Table 1 ijerph-16-04693-t001:** Participants’ background factors.

Participant No.	Female (F)/Male (M)	Age (Years)	Public Sector/Private Sector	No of Employees Participant Was Responsible for
1.	F	59	Public	22
2.	F	Data missing	Private	10
3.	F	47	Private	17
4.	F	62	Public	17
5.	F	51	Public	25
6.	F	43	Public	5
7.	M	53	Public	25
8.	F	59	Public	12
9.	M	59	Public	34
10.	F	60	Public	55
11.	F	42	Private	16
12.	F	58	Public	20
13.	F	60	Public	35
14.	F	54	Public	38
15.	F	63	Private	15
16.	M	57	Private	11
17.	F	39	Public	6
18.	F	61	Public	7
19.	F	58	Public	Data missing
20.	M	61	Public	50
21.	F	49	Public	35
22.	F	41	Public	12

**Table 2 ijerph-16-04693-t002:** Two examples of the analysis process.

Meaning Unit	Condensed Meaning Unit	Code	Sub-Category	Category
He was called AB. He was the senior manager and I was the head of the operation, which was the largest operation. Then there was a unit belonging to another operation. And there, the manager of the unit was a man, K, and it was actually he who made me doubtful when I applied for the job…He was, he had co-workers, but he was not a formal manager. He was in some way responsible for the unit, and the program activities in the house. And then there was the program coordinator who coordinated activities, she was a very opinionated woman who was very strong, so she was a strong informal leader.	Unclear responsibilities, leadership and how the organization was organized. Informal leaders.	Unclear distribution of responsibilities. Informal leadership.	Unclear descriptions of workers’ and managers’ functions, roles and responsibilities	Having unclear roles and responsibilities
After a while, I noticed that there was something in the atmosphere among the employees which did not work, so I interviewed the former manager, then I found out that there were some employees who had been harassing her in various ways. So, she had moved, changed job because of internal problems about her being harassed.	Earlier managers changed job due to internal problems and harassment.	Prior internal problems and conflicts.	Prior unsolved conflicts	Taking over work groups with pre-existing interpersonal conflicts

**Table 3 ijerph-16-04693-t003:** Overview of the results.

Areas	Categories
Factors contributing to the occurrence of bullying	Being new in the managerial position or recently returning from parental or sick leave. Having unclear roles and responsibilities Taking over work groups with pre-existing interpersonal conflicts Being involved in a workplace reorganization
Factors that enabled the bullying to continue	Bystander behavior Higher-level management supporting the perpetrators, bystanders and scapegoating
Strategies used to cope with workplace bullying	Problem-focused coping strategies ◦ Social coping―seeking support from the organization ◦ Social coping―seeking support from colleagues ◦ Confrontation ◦ The power/mandate Emotion-focused coping ◦ Social coping―seeking support from family, friends or social networks ◦ Solitary coping ◦ Avoidance coping Leaving the organization

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
