# Peer review of "Workplace Bullying as Experienced by Managers and How They Cope: A Qualitative Study of Swedish Managers"

_ijerph, 2019, doi:10.3390/ijerph16234693_

Round 1
Reviewer 1 Report
The issue is quite interesting but the presentation of the results is too generic.
The introduction is too long and boring for the reader. It seems to be more a "summa" on the topic than an introduction to the study(lines 36-62 and 77-85). I think that it should be shortened by choosing some few main issues as a background and going quickly to the goals of the study.
Minor: line 27 p.3?; line 53 p.597?; the term "another" is use to frequently (Line 47, 49, ...).
1.1 Theory of coping: i suggest to short the definition and move this paragraph in the methods.
1.2 I suggest to be more clear and precise about all the aims of the study. To what kind of working organization are you dealing with?
2. Methods
2.1 Partecipants:
line 134:please clarify what was the "strategic sampling procedure"
line 138 it is important to report here the definition of "Bullying" that is the main topic addressed.
Lines 140-141 how many work organizations have you considered for the recruitment and how many agreed, the same data are important for the enrolled subjects 27 of ?
Partecipants age: please furnish SD or SE and range.
Table 1 Partecipants n.2 and 19 some data are missing, could you justify in the results and discussion?
Line 167: How long the participants had been working as mangers ?
Interviews: please provide more information on this and quantify how many at home, office or by telephone..... Give SD of time duration.
2.3 Data analysis:
line 188: what it means "qualitative content analysis"? This part is difficult to understand by the non expert readers. Try to semplify and summarize it.
Lines 202-205: what "adjusted after discussion" means? How the "final version" was obtained?
Table 2 reports probably an exemple. Please describe it better in the legend.
It would be great to have results in an additional table based on the exemple of table 2-
Table 3 reports only an overview of the results. Could you provide an additional table with the results of aswers about 3.1. Factors which enable workplace bulling to occur?
The discussion should be revised and more focused on your results.
The methodological consideration should report more extensively the limitations of the study.
Author Response
Thank you very much for all the valueble comments. Please read the revision comments in enclosed file.

Reviewer 2 Report
Thank you for the opportunity to review this manuscript. This study explores an important issue and is well-presented and written. However, my main concern is that the current aim and contribution of this paper are not clearly specified, therefore I am unsure how the findings currently fill a gap or extend our knowledge about the bullying of managers. Aside from this, there are some other key issues with methodology that require further elaboration and a stronger justification. These are listed below:
Major points
The aim of the study “to describe factors related to the workplace bullying…” is unclear. At present, the reader is uncertain whether this is referring to causal/antecedent factors, or coping factors/mechanisms, or other general factors? Ideally this aim(s) would be reworded for clarity. More importantly, it is unclear whether there is an existing gap in the literature that this study is trying to address and what is the specific contribution to theory and/or practice made by the findings of this research. This theoretical gap and contribution also needs to be explicitly established in the introduction section. Likewise, in the Discussion section it is unclear how the findings are additive to our current knowledge about the workplace bullying of managers, beyond the existing literature cited. The extension of knowledge and contribution(s) also needs to be emphasised within this section. A stronger rationale for the chosen qualitative approach is needed in section 2. This involves not necessarily justifying the use of qualitative over quantitative methods, but more about highlighting why a qualitative approach was best suited to address the specific study aim. While the COREQ guidelines are alluded to, further detail is also required in the Method section (section 2) on the following points, to provide context for the reader. Background/rationale for criteria 1 - why did they have to identify as having experienced bullying for ‘approximately 6 months’? Is this duration a conflation with the timeframe in which bullying is usually measured (i.e.; in the past six months)? Perhaps a reference to support this criteria would be beneficial How was the study information distributed to organisations (i.e.; what was the initial sampling method used? Random? Convenience?) More important than the average age of the excluded sub-group (n=5) would be details on why their experience did not fulfil the criteria (i.e.; was this specifically around repetition, duration, or other elements of the definition used?)
Beyond this, addressing the following points would also add further clarity to the arguments presented in this paper.
Minor issues
Suggest avoiding use of gendered language (eg: “his managerial position”, line 42) Lines 55-56 - consider starting section on outcomes of workplace bullying in a separate paragraph (line 55) to the antecedents Line 60-62 specify whether ‘exposure’ means experiencing and/or witnessing bullying - perhaps could also briefly delve into the impact of witnessing bullying, as there is a lot of research on this Line 63 - consider rephrasing to avoid generalisations Lines 97-98 - check wording “The dominant view” ? Lines 108-111 - would be good to elaborate on these two statements (i..e.; emotion-focused coping being ineffective as it exacerbates the stressor-strain relationship, whereas problem-focused coping is more effective) as it is a key tenet of this study (and other workplace bullying research) The ‘methodological considerations’ (lines 459-475) do not appear to be adding anything to the manuscript, and some of the details around data analysis are fairly standard procedures for analysis rather than distinctly adding to the integrity of the findings. Perhaps some of these points might be better integrated into the Method section itself, so that the reader is well-placed to judge of the rigour of the studyAll the best with your research!
Author Response
Thank you very much for all the valuable comments. Please read the revision comments in the enclosed file.

Reviewer 3 Report
Thank you for the opportunity to provide feedback on this manuscript. Overall this manuscript is nicely done.
I recommend moving the ethical considerations section to earlier in the manuscript prior to data collection.
Further, it would be helpful to understand additional detail on the type of organizations where participants worked (healthcare, business, etc.)
please include this information in addition to public/private sector.
Author Response
Thank you for the opportunity to provide feedback on this manuscript. Overall this manuscript is nicely done.
I recommend moving the ethical considerations section to earlier in the manuscript prior to data collection.
Revision response: the section about ethical considerations has been moved to before the section on data collection.
Further, it would be helpful to understand additional detail on the type of organizations where participants worked (healthcare, business, etc.)
please include this information in addition to public/private sector
Revision response: additional information can now be found on lines 161-163.
Round 2
Reviewer 1 Report
The manuscript has been significantly improved. The methods and in particular the criteria of participants recruitment and inclusion criteria are now more clear.
The limitation and the discussion sessions are now adequate.
In my opinion, It would be useful to provide the reader the power (i.e % ? ) of the single factor reported in table 3 in contributing to the occurrence of bullying in the study population.
Author Response
Revision response: Thank you for the possibility to revise and resubmit the manuscript. We agree with you that such information would provide Table 3 (overview of the results) with important information, However, given the nature of the qualitative data collected by means of semi-structured interviews, it’s not possible to provide the percentages of each single factor reported in Table 3. The semi-structured interviews reflected the participants’ experiences of being in a bullying situation and we haven’t included questions related to specific factors. During the interviews, the participants were encouraged to speak about their experiences. The factors were identified with the use of an inductive qualitative method for data analysis.
Best regards
Reviewer 2 Report
This manuscript reflects a carefully considered revision based on the feedback provided, and the merits of this study are more clearly emphasised.
Author Response
Thank you so much. Best regards